# The Evaluation of Industry 5.0 Concepts: Social Network Analysis Approach

Dragana Slavic [1], Ugljesa Marjanovic [1], Nenad Medic [1,2], Nenad Simeunovic [1] and Slavko Rakic [1,*]

1 Faculty of Technical Sciences, University of Novi Sad, Trg Dositeja Obradovica 6, 21000 Novi Sad, Serbia; slavic.draganaa@uns.ac.rs (D.S.); umarjano@uns.ac.rs (U.M.); medic.nenad@uns.ac.rs (N.M.); nsimeun@uns.ac.rs (N.S.)

2 Research and Development Institute for Artificial Intelligence of Serbia, Fruskogorska 1, 21000 Novi Sad, Serbia

* Correspondence: slavkorakic@uns.ac.rs

**Abstract:** During 2022 and 2023, Industry 5.0 attracted a lot of attention. Many articles and papers regarding the basics of Industry 5.0, its pillars, and a comparison of Industry 5.0 and Industry 4.0, Society 5.0, and Operator 5.0 have been published. Although the concept of Industry 5.0 is relatively new, companies from developed countries that have a high level of implementation of Industry 4.0 have already started the transition to Industry 5.0. Even though Industry 5.0 enables developing countries to become a part of developed countries' value chains, it is not known which path to Industry 5.0 developing countries are taking. To fill this gap, the authors proposed research questions regarding the key indicators for measuring the levels of implementation of Industry 5.0 approaches in the manufacturing sector of the Republic of Serbia. This research includes insights from 146 manufacturing companies, gathered in 2022 as a part of the European Manufacturing Survey. The main findings of this study show that the most important indicator when it comes to human-centricity is training and competence development of production employees with a task-specific focus; the implementation of measures for improving efficiency in material consumption is significant for achieving sustainability; and the use of standardized and detailed work instructions is crucial in order to become resilient.

**Keywords:** Industry 4.0; Industry 5.0; Social Network Analysis; European Manufacturing Survey

## 1. Introduction

Industry 5.0 represents a strategy designed for filling in major Industry 4.0 deficiencies, by combining the technologies developed in Industry 4.0 with Society 5.0 and the Operator 5.0 principles [1–3]. These deficiencies are mainly related to neglecting people both as a crucial part of many kinds of processes and as customers [3]. Industry 5.0 aims to use the Industry 4.0 technologies, such as the internet of things, big data, and collaborative robots, for meeting goals, which propose a better future for society as a whole, and for humans as individuals [1,3–5]. The aforementioned goals are meant for fulfilling the Sustainable Development Goals created by the United Nations and for designing improved workplaces for employees [2,4]. Accordingly, Industry 5.0 requires the use of Industry 4.0 technologies in a way that will make new opportunities for supporting the main ideas of Society 5.0 and Operator 5.0 [1,2]. Society 5.0 presents a vision of a society that uses new-age technologies for solving crucial societal problems, such as hunger, poverty, and environmental issues [6]. When it comes to Operator 5.0, it is crucial to understand how workplaces have to be organized in order to provide successful collaboration between humans and robots [1,4].

Besides bringing the focus back to human touch and people in general, sustainability and resilience are being addressed by Industry 5.0 as well [7]. This led to the three main pillars of Industry 5.0: human-centricity, sustainability, and resilience [8]. Researching Industry 5.0 became a trend in 2022, and hundreds of articles and papers on a wide range

of topics have been written: what is Industry 5.0, how and when was it initiated, what are its pillars, what are the practical implications, how can Industry 5.0 be implemented, what are its benefits, and how does it compare with Industry 4.0. Slavic et al. described the main concepts of Industry 5.0 [7], Madsen et al. described the birth and emergence of Industry 5.0 [9], Cockalo and collaborators showed the transition process to Society 5.0 [10], and Xavier et al. showed the bibliometric analysis of the Industry 5.0 development [11].

The previously mentioned Industry 5.0 pillars—human-centricity, sustainability, and resilience—have found their place in already developed countries, which are in the process of transitioning from Industry 4.0 to Industry 5.0 [12]. In addition to developed countries, Industry 5.0 is of great significance for developing countries as well, because it provides new opportunities for establishing value chains and improving the situation of these countries in general [5,13]. Furthermore, for successful Industry 5.0 implementation and keeping its position in the market, it is crucial to do business according to SDGs. Since the majority of Industry 5.0 studies examine the strategy itself, a gap is identified. This gap refers to: (1) the practical implementation of Industry 5.0 in firms; (2) which indicators can be used to measure the implementation level of Industry 5.0 approaches; and (3) how to measure the level of implementation. The previously mentioned papers and articles address Industry 5.0 in a theoretical manner. Javaid et al. addressed the potential practical application of Industry 5.0 in medicine [14], and Al-Emran et al. discussed the future implementation of Industry 5.0 in education, emerging in Education 5.0 [15]. Nevertheless, research on the current application of Industry 5.0 in the manufacturing sector of developing countries is lacking. In order to give insights on the state of Industry 5.0 in developing countries, as well as improvement suggestions, the authors propose the following research questions:

RQ1: *Which are the key indicators for measuring the level of implementation of human-centricity in the manufacturing sector in developing countries?*

RQ2: *Which are the key indicators for measuring the level of implementation of sustainability in the manufacturing sector in developing countries?*

RQ3: *Which are the key indicators for measuring the level of implementation of resilience in the manufacturing sector in developing countries?*

According to the research questions, this paper contributes by acknowledging the Industry 5.0 pillars individually and providing information on the indicators of each pillar, as well as data on the current state of the pillars in the Republic of Serbia, one of the developing countries. Research has shown that human-centricity is achieved by investing in production employees' training and competence development with a task-specific focus, as well as by establishing bonus systems for outstanding performances in production and/or innovation. When talking about sustainability, developing countries are achieving this by implementing certain measures for improving efficiency in material consumption and energy use; possessing a certified environmental management system is the third most important factor in achieving sustainability. The use of standardized and detailed work instructions has the biggest contribution to becoming and staying resilient, followed by the implementation of activities that raise employees' awareness of data security and the use of software specifically as two important factors for resilience.

To successfully answer the research questions and to give insights on the manufacturing sector of the Republic of Serbia, the structure of this paper includes an introduction in Section 1, a literature review covering Digital Product-Service Systems and three Industry 5.0 approaches in Section 2, methodology with information on data collection and data analysis in Section 3, results and discussion in Section 4, and a conclusion in Section 5.

## 2. Literature Review

### 2.1. Digital Product-Service Systems

When firms started to offer services and products together, the term Servitization was created [16]. During the previous years, firms have started to use Servitization as a strategy for doing business, which has triggered the appearance of Product-Service Systems [17,18]. Product-Service Systems represent a way to increase a product's value,

by including different kinds of services as a part of the final version of the product that is offered to the end users [19,20]. These services range from basic to advanced, where the basic ones represent services related to the characteristics of the product, while the advanced services do not relate to the product's characteristics [21].

The initiation of Industry 4.0, known as the Fourth Industrial Revolution, has caused Servitization to grow into Digital Servitization, and Product-Service Systems have a new, digital dimension—Digital Product-Service Systems [16,22]. When offering Digital Product-Service Systems, firms are including digital services with the products aimed for end users [23].

In the context of Industry 5.0, Digital Product-Service Systems contribute to all three of the main approaches [24]. The contribution to human-centricity reflects in the number of people needed to successfully implement and manage the Digital Product-Service Systems, which enables improvement of the existing value chains and the creation of new ones [25,26].

Additionally, Digital Product-Service Systems support sustainability because of their digital aspects—not only do they influence business models in a manner that makes them long term and sustainable, but they also lower the negative impact that firms are having on the environment, both locally and globally [27,28].

Finally, during the COVID-19 pandemic, Digital Product-Service Systems proved to be resilient, precisely because of their digital aspects [24]. Digital services can be provided no matter the time and place, the main condition is to have Internet access so that the digital services can be delivered [29].

### 2.2. Industry 5.0: Human-Centric Approach

Industry 4.0 caused the development of many new technologies—autonomous robots, simulations, horizontal and vertical system integrations, the internet of things (IoT), cybersecurity, cloud computing, additive manufacturing, augmented reality and virtual reality (AR and VR), and big data and analytics [30,31]. In the context of the Fourth industrial revolution, these technologies are used to achieve mass product customization and a higher level of automatization and digitalization, as well as better coordination of cyber-physical systems [32]. These achievements have been made, and one big deficiency was identified—humans were removed from many workplaces and replaced by robots and machines, which has caused the shutdown of many jobs [33]. Furthermore, while Industry 4.0's main focus is to make a higher profit by lowering the costs in all business spheres, Industry 5.0 requires firms to think about the global environment during the decision-making process [3]. This requirement arises from the previously mentioned SDGs, which are found in all three Industry 5.0 approaches [34,35]. In order to fulfill this requirement, it is necessary to include SDGs and their values in education, with an aim to raise awareness about this topic. Additionally, Education 5.0 will prepare pupils and students for future jobs that imply human–robot collaboration, by teaching them how to co-exist and manage the work being performed with the help of collaborative robots [36,37].

In the context of human-centricity, SDGs and Society 5.0 contribute by respecting employees, offering better working conditions, and observing people as a part of the value chain [6]. Human-centricity can be observed in two directions—the operators, and the customers [16].

When it comes to the operators, Operator 5.0 is a concept that has the biggest impact on defining the standards for new workplaces for manufacturing companies [4]. Future workplaces should consider using collaborative robots, which will work together with people (operators) during the execution of different tasks [1]. During the task execution, robots will be focused on dangerous, repetitive tasks, while humans will contribute through cognitive and creative work [16,38]. This work approach transforms manufacturing into mindfacturing, meaning that the main production factors are knowledge and talent, which enables the mass personalization of products [39]. Mass personalization is one of the main factors that differentiates Industry 5.0 from Industry 4.0 [3]. Some of the ways in which

robots and humans will collaborate are gesture recognition and intention prediction, stress monitoring, and cognitive abilities development [33].

From the perspective of customers' involvement, Industry 5.0 emphasizes the inclusion of the customers and end users in the value chain. This is achieved by including them in the production process, which results in a higher level of product personalization and stronger company–customer connection and loyalty [5].

According to the literature, human-centricity is measured by investments made in effective communication, unit empowerment, and personal growth [40]. Correspondingly, parameters used for measuring human-centricity in this research are: (1) the use of interactive interfaces with the operator; (2) the use of an internet/network connection in real time for automated data exchange; (3) the integration of tasks; (4) employee involvement in innovation development; (5) employee bonus systems for outstanding performances in production and/or innovation; (6) training and competence development of production employees with a task-specific focus; (7) training and competence development of production employees with a cross-functional focus; (8) training and competence development of production employees to support the implementation and use of digital production technologies or digital assistance systems; (9) training and competence development of production employees in data security and data compliance; and (10) training and competence development of production employees in creativity and innovation.

### 2.3. Industry 5.0: Sustainability Approach

Considering sustainability issues, three types of factor appear—social, economic, and ecological [41]. Social factors are related to the society and its needs, while economic factors reflect the economic steadiness of a particular firm or industry, and ecological factors represent the state of the environment [27,41,42]. These factors are also referred to throughout the Sustainable Development Goals (SDGs) established by the United Nations [6].

Industry 4.0 addressed sustainability, but only partially on a micro-organizational level [43]. Industry 5.0 approaches sustainability in a broader manner, taking into consideration not only issues on a micro-organizational level, but on a macro-organizational level, and on a global level as well [44].

It is expected that firms will develop corporate social responsibility, which supports informing and educating employees about current social and environmental problems, and encourages them to find new ways of doing business that have a weak or no negative influence on the environment, through implementing the use of new materials, recycling the products or their parts, etc. [1,6,45]. It is also expected that production planning will be performed according to planet Earth's possibilities and constraints, because an increase in the production volume leads to an increase in gas emissions [1,46].

Additionally, when purchasing goods, modern customers seek to support businesses that not only offer high-quality products but also align with their values—this trend has led to an increasing emphasis on green business and the circular economy [35,47].

In this research, the parameters used for measuring sustainability were: (1) the possession of a certified environmental management system; (2) the possession of a certified energy management system; (3) the implementation of measures for improving efficiency in material consumption; (4) the implementation of measures for improving efficiency in energy use; (5) the implementation of measures for improving efficiency in water use; (6) the use of technologies for recycling and re-use of water; (7) the use of technologies to recuperate kinetic and process energy; (8) the use of technology that resulted in a significantly more efficient use of materials when first implemented; (9) product revamping or modernization; and (10) offering take-back services.

### 2.4. Industry 5.0: Resilience Approach

As a term related to firms and industry, resilience started to gain attention in 2020, during the COVID-19 pandemic [7]. The challenges that firms faced during this period required them to adapt their business models to newly established circumstances and

continue doing business as usual, as well as to go back to their original business models when the circumstances changed to their previous state [48].

Resilience is improved by implementing digital solutions that contribute to different aspects regarding growth and development, as well as quicker responses to problems [49]. In the context of Industry 5.0, it is expected that firms, supply chains, processes, and business models will become resilient on a daily basis, in order to successfully address both the positive and negative impacts of their environment [35,47]. Resilience can be achieved by using adaptable technologies, which can improve the quality of the final products [2,50]. These technologies were developed during Industry 4.0, and will continue to be used in Industry 5.0. The most used technologies are the internet of things and artificial intelligence [29]. Additionally, technologies that will increase firms' resilience and are expected to be used more frequently are collaborative robots, as well as digital twins [4,51].

Resilience can be established on an industry level by digitalizing the economy and creating a partnering network between firms, which would bring collaboration to its full potential [50]. Digitalizing the economy by implementing circular economy means that business models are becoming sustainable and more valuable, which contributes to the Industry 5.0 sustainability approach as well [30,52]. Additionally, firms can improve resilience and sustainability by using renewable energy sources that will contribute to the whole industry [50]. These changes are also significant for the human-centric approach, because they impact the customer's decision when buying products, and they can improve customer loyalty [5,45].

The parameters used for measuring resilience in this research are: (1) the use of an internet/network connection in real time for automated data exchange; (2) the use of standardized and detailed work instructions; (3) the use of mobile industrial robots; (4) the use of collaborating robots (co-bots); (5) the implementation of activities raising employees' awareness of data security; (6) the use of software specifically; (7) the use of hardware solutions specifically; (8) the use of organizational measures specifically; and (9) product revamping or modernization.

## 3. Methodology

### 3.1. Data Collection

The dataset for this research was obtained in 2022, as a part of the European Manufacturing Survey (EMS). The data collection process was performed in four stages: (1) collecting information about the manufacturing companies in the Republic of Serbia; (2) contacting the companies; (3) sending the survey; and (4) gathering that data. The authors chose the Republic of Serbia to represent developing countries due to the previous literature in the manufacturing sector, which identified this country as a notable example within the research consortium of the European Manufacturing Survey. The survey was coordinated by the Fraunhofer ISI from Germany.

The Serbian industry consists of 2489 manufacturing firms—4% are micro firms, meaning that they have less than 20 employees, 62% are small firms with 20 to 49 employees, 26% are medium-sized firms with 50 to 249 employees, while the rest (8%) are large firms with more than 250 employees. The five most common manufacturing sectors are the manufacture of food products (19.6%); the manufacture of fabricated metal products, except machinery and equipment (14.5%); the manufacture of rubber and plastic products (7.6%); the manufacture of wearing apparel (7.2%); and the manufacture of machinery and equipment n. e. c. (5.3%). As well as being grouped by their size and manufacturing sector, the firms are also grouped by the district they belong to—21.3% are from the Belgrade district, 9.9% are from the South Backa district, 7.6% are from the Srem district, 5.6% are from the Zlatibor district, and 5.3% are from the Morava district.

According to the Serbian population and the consortium's guidelines, a sample for this research was created. The sample included 800 firms in total—56% were small firms, 30% were medium firms, and 10% were large firms. The remaining 4% were micro firms, which

could not take part in the research, according to the guidelines. The five most common manufacturing sectors in this sample were the manufacture of food products (19%); the manufacture of fabricated metal products, except machinery and equipment (14.1%); the manufacture of rubber and plastic products (7.8%); the manufacture of wearing apparel (7.4%); and the manufacture of machinery and equipment n. e. c. (5%). The five most common Serbian districts in the sample were Belgrade (18%), South Backa (9.8%), Srem (6.9%), Zlatibor (5.9%), and Morava (5.6%).

The companies that could take part in this survey must be manufacturing companies that have more than 20 employees. In order to satisfy the criteria, the Agency for Business Registers in the Republic of Serbia has given a dataset that provides all manufacturing companies with more than 20 employees founded in the Republic of Serbia.

Secondly, the firms were contacted by phone calls. The phone calls were directed to the production managers or CEOs of the companies, since they were the ones who should fill in the survey. During the phone call, more information about EMS was given so that the firm's representative could decide to give consent for taking part in the survey.

The next stage required sending the survey to all the firms that agreed to take part. The first round of the sending process was conducted the following day after obtaining consent. If the answer has not been received in a period of two weeks, the second round of the sending process was performed, when reminders for filling in the survey were sent.

Finally, after collecting all the answers, data from the surveys were gathered in a spreadsheet, which was followed by a deeper analysis.

The final sample for this research gathered data from 146 firms. These firms were divided into three groups—small firms that have from 20 to 49 employees, medium firms that have 50 to 249 employees, and big firms that have more than 250 employees, presented in Table 1. This firm size distribution scale is based on manufacturing innovations, defined by the European Manufacturing Survey's consortium [53]. Additionally, this distribution scale was used in previous EMS-related papers and articles, and was proven to be appropriate [21,53–56].

**Table 1.** Sample distribution by firm size.

| Firm Size | n | % |
|---|---|---|
| Small | 65 | 45 |
| Medium | 57 | 39 |
| Large | 24 | 24 |
| Total | 146 | 100 |

As shown in Table 1, 65 small firms (45%), 57 medium firms (39%), and 24 large firms (16%) from Serbia took part in EMS 2022.

Additionally, the firms were differentiated by the NACE classification, presented in Table 2. This classification groups firms by their business.

**Table 2.** Sample distribution by business sector classification.

| Manufacturing Industry | Share of the Total Sample (n) | Share of the Total Sample (%) |
|---|---|---|
| Manufacture of fabricated metal products, except machinery and equipment | 28 | 19 |
| Manufacture of food products | 24 | 16 |
| Manufacture of rubber and plastic products | 12 | 8 |
| Manufacture of wearing apparel | 9 | 6 |
| Manufacture of machinery and equipment n. e. c. | 9 | 6 |
| Manufacture of non-metallic mineral products | 8 | 5 |
| Manufacture of basic metals | 8 | 5 |
| Other | 48 | 35 |
| Total | 146 | 100 |

While conducting the research, firms from all NACE business sectors were contacted. During the data analysis process, business sectors with a significant share in the total sample were singled out, because they represented the current state of the manufacturing sector of the Republic of Serbia. These sectors are shown in Table 2, while the rest of the NACE business classes were grouped in "Other". NACE business classes that were grouped in "Other" had less than 8 firms per class, totalling 48 firms.

As presented in Table 2, the manufacture of fabricated products, except machinery and equipment (NACE 25) had 19% of the total sample (28 firms), followed by the manufacture of food products (NACE 10), which had 16% of the total sample (24 firms). The manufacture of rubber and plastic products (NACE 22) accounted for 8% of the total sample (12 firms), while the manufacture of wearing apparel (NACE 14) and the manufacture of machinery and equipment n. e. c. (NACE 28) had 6% (9 firms) each. The manufacture of non-metallic mineral products (NACE 23) and the manufacture of basic metals (NACE 24) had 5% of the total sample (8 firms) each. Finally, the remaining 35% of the total sample (48 firms) belonged to the other manufacturing sectors.

*3.2. Data Analysis*

In this research, the data were analyzed using the Social Network Analysis (SNA) method, which shows the bond strength between different actors in the same network. The networks in this research represent the three main Industry 5.0 pillars—human-centricity, sustainability, and resilience. Additionally, the actors are the firms that took part in the survey and the aforementioned parameters, which make two-mode networks. The networks show the existing relationships between the firms and parameters, and detect which parameters are usually used in the firms.

Data visualization was performed using graphs that show red circles and blue squares. The circles represent firms that were a part of the research, while the squares represent parameters. The firms and parameters were both labelled using a combination of letters and numbers. The firms were labelled from F1 to F146, the human-centric parameters were labelled from HC1 to HC10, the sustainability parameters from were labelled S1 to S10, and the resilience parameters were labelled from R1 to R9. Additionally, the parameters were ranked using degree, closeness, betweenness, and eigenvector.

The human-centric parameters used in this research were: HC1—the use of interactive interfaces with the operator; HC2—the use of an internet/network connection in real time for automated data exchange; HC3—the integration of tasks; HC4—employee involvement in innovation development; HC5—employee bonus systems for outstanding performances in production and/or innovation; HC6—training and competence development of production employees with a task-specific focus; HC7—training and competence development of production employees with a cross-functional focus; HC8—training and competence development of production employees to support the implementation and use of digital production technologies or digital assistance systems; HC9—training and competence development of production employees in data security and data compliance; and HC10—training and competence development of production employees in creativity and innovation.

The sustainability parameters used in this research were: S1—the possession of a certified environmental management system; S2—the possession of a certified energy management system; S3—the implementation of measures for improving efficiency in material consumption; S4—the implementation of measures for improving efficiency in energy use; S5—the implementation of measures for improving efficiency in water use; S6—the use of technologies for recycling and re-use of water; S7—the use of technologies to recuperate kinetic and process energy; S8—the use of technology, which resulted in a significantly more efficient use of materials when first implemented; S9—product revamping or modernization; and S10—offering take-back services.

The resilience parameters used in this research were: R1—the use of internet/network connection in real time for automated data exchange; R2—the use of standardized and

detailed work instructions; R3—the use of mobile industrial robots; R4—the use of collaborating robots (co-bots); R5—the implementation of activities raising employees' awareness of data security; R6—the use of software specifically; R7—the use of hardware solutions specifically; R8—the use of organizational measures specifically; and R9—product revamping or modernization.

## 4. Results and Discussion

### 4.1. Social Network Analysis

Tables 3–5 show the results of the centrality between firms. The results are shown through degree, closeness, betweenness, and eigenvector. Degree gives information about the importance of an actor in the network—the higher the degree, the greater the importance. Closeness represents the position of the actor in the network when compared with other actors. Betweenness measures if a certain actor lies on some of the nodes created in the network. Finally, eigenvector shows how the most important actors of the network are interconnected. Additionally, Figures 1–3 show the networks for the aforementioned Industry 5.0 approaches. In these figures, the red dots represent the firms that took part in the research, while the blue squares represent the parameters corresponding to the approach.

**Table 3.** Results of centrality for the human-centric approach.

| Parameter | Degree | Closeness | Betweenness | Eigenvector |
|-----------|--------|-----------|-------------|-------------|
| HC1 | 0.014 | 0.308 | 0.012 | 0.006 |
| HC2 | 0.260 | 0.383 | 0.079 | 0.168 |
| HC3 | 0.315 | 0.394 | 0.044 | 0.238 |
| HC4 | 0.527 | 0.469 | 0.120 | 0.392 |
| HC5 | 0.568 | 0.485 | 0.162 | 0.405 |
| HC6 | 0.753 | 0.569 | 0.334 | 0.485 |
| HC7 | 0.356 | 0.410 | 0.041 | 0.292 |
| HC8 | 0.404 | 0.425 | 0.051 | 0.327 |
| HC9 | 0.397 | 0.423 | 0.048 | 0.322 |
| HC10 | 0.301 | 0.394 | 0.024 | 0.257 |

**Table 4.** Results of centrality for the sustainability approach.

| Parameter | Degree | Closeness | Betweenness | Eigenvector |
|-----------|--------|-----------|-------------|-------------|
| S1 | 0.397 | 0.432 | 0.180 | −0.405 |
| S2 | 0.110 | 0.353 | 0.009 | −0.113 |
| S3 | 0.500 | 0.469 | 0.250 | −0.513 |
| S4 | 0.459 | 0.453 | 0.174 | −0.512 |
| S5 | 0.233 | 0.383 | 0.035 | −0.287 |
| S6 | 0.137 | 0.360 | 0.022 | −0.155 |
| S7 | 0.144 | 0.361 | 0.017 | −0.157 |
| S8 | 0.356 | 0.418 | 0.151 | −0.351 |
| S9 | 0.185 | 0.371 | 0.060 | −0.163 |
| S10 | 0.130 | 0.358 | 0.027 | −0.133 |

**Table 5.** Results of centrality for the resilience approach.

| Parameter | Degree | Closeness | Betweenness | Eigenvector |
|-----------|--------|-----------|-------------|-------------|
| R1 | 0.260 | 0.371 | 0.075 | −0.214 |
| R2 | 0.637 | 0.498 | 0.308 | −0.545 |
| R3 | 0.007 | 0.311 | 0.000 | −0.009 |
| R4 | 0.034 | 0.321 | 0.001 | −0.031 |
| R5 | 0.610 | 0.486 | 0.263 | −0.545 |
| R6 | 0.432 | 0.421 | 0.110 | −0.423 |
| R7 | 0.356 | 0.398 | 0.070 | −0.361 |
| R8 | 0.123 | 0.341 | 0.006 | −0.133 |
| R9 | 0.185 | 0.353 | 0.016 | −0.180 |

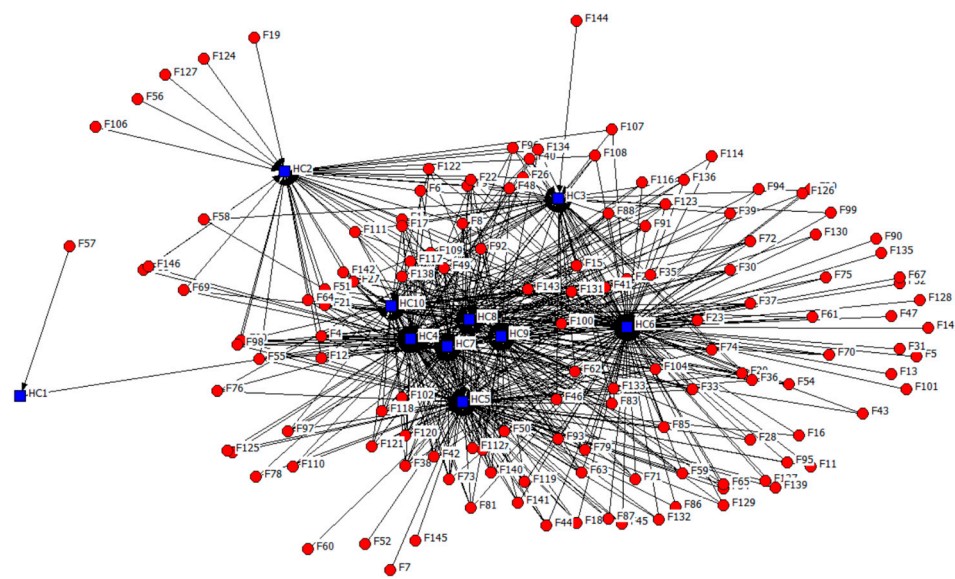

**Figure 1.** Social network for the human-centric approach.

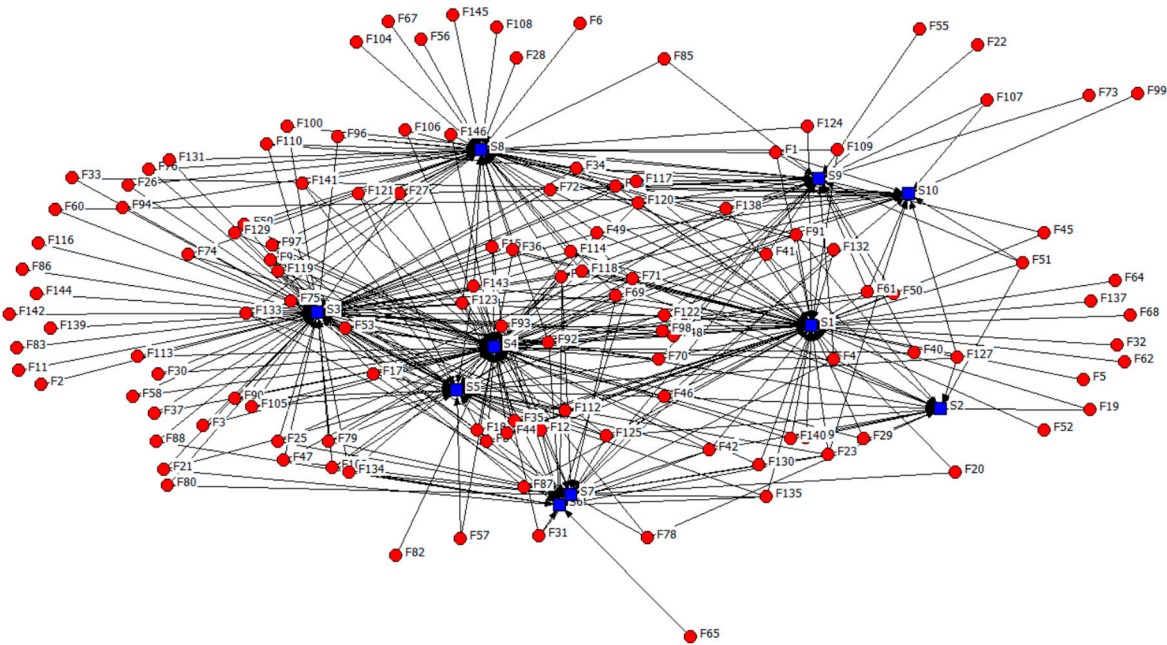

**Figure 2.** Social network for the sustainability approach.

Table 3 gives information about the human-centric approach.

As shown in Table 3, 10 human-centric parameters were analyzed. The most important factor for achieving human-centricity was HC6 (training and competence development of production employees with a task-specific focus). HC6 is also the closest to other actors in the network; it lies on several network nodes, and it is the most connected with other important actors of the network. HC6 was followed by HC5 (employee bonus systems for outstanding performances in production and/or innovation), which is the second most important actor in the network, meaning that it is usually found in Serbian manufacturing firms as a way to achieve human-centricity. This parameter is significantly connected to other parameters, but it is found on a minority of network nodes. Also, its eigenvector score is significant, because this parameter is connected to other important parameters, such as the previously mentioned HC6. The third most significant actor was HC4 (employee involvement in innovation development), which also had a good connection to the other

actors, resulting in a relatively good eigenvector score; however, HC4′s betweenness had a low score, which means that fulfilling this parameter does not affect the fulfilment of other actors of the same network. According to degree score, HC4 was followed by HC8 (training and competence development of production employees to support the implementation and use of digital production technologies or digital assistance systems). Compared with the abovementioned parameters, this parameter had lower, but significant, closeness and eigenvector scores. Additionally, its betweenness score was low, meaning that it did not affect the achievement of other actors of the network. HC5 (training and competence development of production employees in data security and data compliance) was the fifth most significant human-centric indicator. Although this indicator did not influence other actors in the network, it was close to them, which was seen in both the closeness and eigenvector scores. HC7 (training and competence development of production employees with a cross-functional focus) was not crucial for establishing human-centricity in Serbian manufacturing firms; however, this indicator was close to the other indicators related to training and competence development, meaning that they partially influenced each other. When it comes to improved ways of working, HC3 (integration of tasks) showed that this indicator was not common among the firms that took part in this research, resulting in a low degree score. This also caused low closeness to other network actors, as well as low betweenness and eigenvector scores. HC3 was followed by HC10 (training and competence development of production employees towards creativity and innovation). This actor also had low degree, closeness, betweenness, and eigenvector scores since these kinds of training and competence development are not usually held. Next, HC2 (the use of internet/network connection in real time for automated data exchange) was not often implemented in Serbian manufacturing firms, which was followed by low degree, closeness, betweenness, and eigenvector scores. Finally, HC1 (the use of interactive interfaces with the operator) was rarely found, which impacts its betweenness score, meaning that it usually does not lie on the nodes of the network. Closeness and eigenvector values were also the lowest, when compared with other network actors. These data are visually presented in Figure 1.

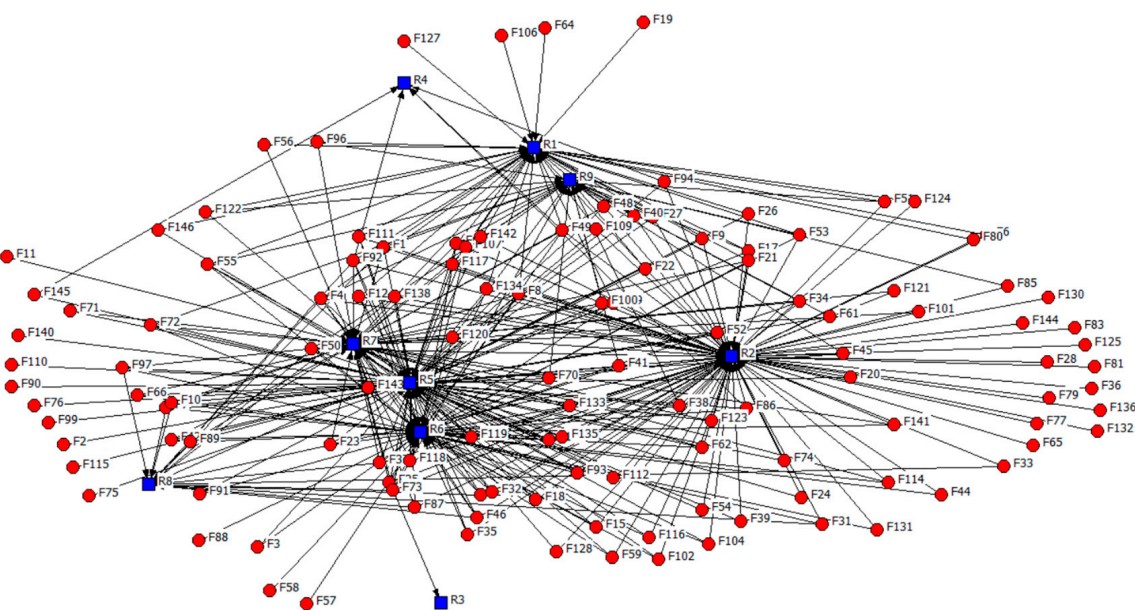

**Figure 3.** Social network for the resilience approach.

Table 4 gives information about the sustainability approach.

The parameters, and the degree, closeness, betweenness, and eigenvector scores of the sustainability approach are given in Table 4. The most influential actor of the network was definitely S3 (implementation of measures for improving efficiency in material

consumption), meaning it was usually found in Serbian manufacturing firms. This indicator had a good connection to other actors in the network, as it lies on several network nodes, but it was not well connected to the other important actors of the same network. The second most important indicator for sustainability was S4 (implementation of measures for improving efficiency in energy use). Similar to S3, S4 had a relatively good closeness value, but its betweenness value was significantly lower. Its eigenvector value showed that this indicator does not have a good connection with the other important network actors. S1 (possession of a certified environmental management system) showed that this is one of the crucial indicators for achieving sustainability, which was also seen in its closeness and betweenness scores. However, even though its eigenvector score was higher when compared with the previously mentioned sustainability parameters, S1 also is not well connected to the important actors of the network. S8 (the use of technology, which resulted in a significantly more efficient use of materials when first implemented) was the fourth most important actor of the sustainability network. Its closeness and betweenness scores were relatively good, and its eigenvector value showed improvement when compared with S3, S2, and S1. S8 was followed by S5 (implementation of measures for improving efficiency in water use). The closeness and betweenness scores of this indicator showed that its influence on other actors of the network was less significant, but the eigenvector score showed that it had a better connection to important sustainability actors. S9 (product revamping or modernization) was not usually found in Serbian manufacturing firms. Similar to S5, S9 had lower closeness and betweenness values, but a higher eigenvector value. S9 was followed by S7 (the use of technologies to recuperate kinetic and process energy), and S6 (the use of technologies for recycling and re-use of water), which showed that the technologies were rarely used for managing energy and water use. This was reflected in the degree, closeness, betweenness, and eigenvector values. S10 (offering take-back services) was also rare in manufacturing firms in the Republic of Serbia, which was seen in low degree, closeness, and betweenness scores; however, this indicator had a better connection to important network actors, which was reflected in the eigenvector score. Lastly, S2 (possession of a certified energy management system) was the least influential actor of the network. Its connection to other actors was not good, and it did not lie on network nodes, but its eigenvector score was the highest. Interestingly, even though sustainability had important factors, they were not interconnected. These data are visually presented in Figure 2.

Table 5 shows the results of centrality for the resilience approach.

Table 5 shows the resilience indicators. Indicators' degrees significantly varied, and their eigenvector values were below 0. The most influential actor of the network was R2 (the use of standardized and detailed work instructions), meaning it was common in Serbian manufacturing firms. This parameter had significant closeness and betweenness values, but its eigenvector score was the lowest, meaning that it did not have a good connection to the other important indicators. The second most important resilience indicator was R5 (implementation of activities raising employees' awareness of data security), which had slightly lower closeness and betweenness scores than R2, and the same eigenvector value. R5 was followed by R6 (the use of software specifically). Even though this indicator had relatively good degree and closeness values, the betweenness value shows that it did not lie on the network nodes; however, its connection with important network actors was better when compared with previously mentioned resilience parameters. The next indicator was R7 (the use of hardware solutions specifically), which showed similar results to R6—despite having relatively good degree and closeness values, its betweenness value was low, but it had a better connection to the important network actors. R1 (the use of internet/network connection in real time for automated data exchange) was also rarely found in Serbian manufacturing firms, resulting in lower closeness and betweenness values. R9 (product revamping or modernization) was not often implemented, but it had a relatively good connection to the other actors in the network, as well as the important ones. On the other hand, its betweenness value was not significant, meaning that it lies on a minority of the

resilience network nodes. R8 (the use of organizational measures specifically) showed that firms that took part in the research did not specify the use of organizational measures, even though this indicator can impact the implementation and improvement of other resilience indicators. Lastly, R4 (the use of collaborating robots (co-bots)) and R3 (the use of mobile industrial robots) indicated the use of robots in manufacturing. This use did not significantly influence the other resilience indicators, which resulted in low degree, closeness, and betweenness values. Figure 3 presents these data visually.

### 4.2. Theorethical and Practical Implications

The results from this study confirm previous research that focused on the difference between Industry 5.0 and Industry 4.0 technologies [2]. Further, these results show the importance of the SDGSs nominated by the European Commission [2,35,57,58]. These facts have brought up a question on whether Industry 5.0 is a revolution or just an evolution [3]. Even though Industry 5.0 does not bring many changes on the technological side, it brings radical changes on the managerial side, meaning that it changes the factors that have a crucial role in the decision-making process. Some of these factors are employees and their well-being; consumers, customers, and their needs; the environment, and the changes certain decisions will have regarding pollution and different gas emissions; as well as improving risk management in order to become resilient and to maintain resilience.

All the parameters used in this research, as well as being divided by Industry 5.0 approaches, can be divided by their level of intensity into a high, medium, and low level of intensity.

The human-centric parameters that have a high level of intensity are training and competence development of production employees with a task-specific focus, employee bonus systems for outstanding performances in production and/or innovation, and employee involvement in innovation development. Not only are these activities common in the firms that took part in this research, but they also influence the other human-centric parameters in a way that supports their development and implementation. The human-centric parameters that have a medium level of intensity are training and competence development of production employees with a cross-functional focus, training and competence development of production employees to support the implementation and use of digital production technologies or digital assistance systems, and training and competence development of production employees in data security and data compliance. Although these activities are not as frequent as the previously mentioned ones, they are significant for carrying out the human-centric approach. Additionally, their implementation can be easier if the high-intensity level parameters are already used in the firms. The low-intensity level human-centric parameters are the use of interactive interfaces with the operator, the use of internet/network connection in real time for automated data exchange, the integration of tasks, and training and competence development of production employees towards creativity and innovation. These activities are not common in the manufacturing sector in developing countries, and their implementation requires relatively big financial investments.

High-intensity level sustainability indicators are the possession of a certified environmental management system, the implementation of measures for improving efficiency in material consumption, and the implementation of measures for improving efficiency in energy use. They are common in the manufacturing sector in developing countries because they contribute to the efficiency of the company, as well as to its reputation. On the contrary, even though they are significantly used in firms, these parameters are not the ones that lead to easier implementation of the other sustainability parameters. The possession of a certified energy management system, offering take-back services, and the use of technologies for the recycling and re-use of water are the ones that contribute to the implementation of medium-intensity level and low-intensity level parameters of this approach. Although they belong to low-intensity level indicators, along with the use of technologies to recuperate kinetic and process energy, and product revamping or

modernization, they are the key for carrying out sustainability. Medium-intensity level sustainability parameters are the implementation of measures for improving efficiency in water use and the use of technology, which resulted in a significantly more efficient use of materials when first implemented. This means that they are easily implemented when the high-intensity level parameters are already used in firms.

When it comes to resilience high-intensity level parameters, they are the use of standardized and detailed work instructions, the implementation of activities raising employees' awareness on data security, and the use of software specifically. They are the most common among firms in the manufacturing sector in developing countries, but they are not key for carrying out resilience. The indicators that are crucial for establishing and maintaining resilience belong to the low-intensity level, and they are the use of mobile industrial robots, the use of collaborating robots (co-bots), and the use of organizational measures specifically. Although they are not common, they are an important factor for achieving resilience. Medium-intensity level parameters of this approach are the use of internet/network connection in real time for automated data exchange, the use of hardware solutions specifically, and product revamping or modernization. These activities contribute to resilience, but they are not relatively easy to implement.

According to the analysis performed by the authors, in order to strengthen Industry 5.0, firms should make investments that will have long-term benefits for the firm itself, but for society as well. These investments include: (1) new learning programmes and opportunities for employees; (2) data centralization, (3) task integration for human-centricity; (4) developing energy management systems; (5) designing and implementing take-back services as a part of the firm's offer in the market; and (6) implementing technologies for recycling and water re-use in the production systems for sustainability. For resilience, it is necessary to: (1) include industrial robots as a part of production systems; (2) include co-bots in workplaces and develop processes that require human–robot collaboration; and (3) centralize data and enable its real time use.

## 5. Conclusions

As a concept, Industry 5.0 has attracted a lot of attention in the previous years. Many authors have defined and discussed the term, as well as its approaches—human-centricity, sustainability, and resilience. Also, some authors have conducted bibliometric analyses. A minority of authors have decided to write about the practical applications of Industry 5.0 in production systems.

After analyzing the answers from 146 manufacturing firms in the Republic of Serbia, the authors of this paper have found the key indicators for measuring the level of implementation of human-centricity, sustainability, and resilience in manufacturing firms in developing countries.

When measuring human-centricity, the key indicators are training and competence development of production employees with a task-specific focus, employee bonus systems for outstanding performances in production and/or innovation, and employee involvement in innovation development, which are also necessary in order to implement the other activities that impact human-centricity positively.

Significant parameters that indicate good levels of sustainability are the possession of a certified environmental management system, the implementation of measures for improving efficiency in material consumption, and the implementation of measures for improving efficiency in energy use, but they are not key factors for maintaining sustainability. In order to be sustainable in the long term, it is necessary to have a certified energy management system, offer take-back services, and use technologies for recycling and re-use of water.

Long-term business is supported by resilience. Key resilience indicators are the use of standardized and detailed work instructions, the implementation of activities raising employees' awareness of data security, and the use of software specifically. To achieve long-term resilience, investments in industrial robots, collaborative robots, and specific organizational measures must be made.

The main limitation of this paper is the data used in the research, since it was gathered only from the manufacturing sector of the Republic of Serbia. Future research should include data from other developing countries, as well as from other manufacturing industries. Additionally, this research gives a general perspective of the manufacturing sector. Future research could focus on manufacturing sectors individually, meaning that it could analyze the Industry 5.0 implementation in a certain manufacturing sector from the NACE classification. Future research should also compare action plans made by developing countries with action plans made by developed countries, in the context of Industry 5.0, and create a joint value chain that gives opportunities for improvement of all value-chain actors.

**Author Contributions:** Conceptualization, N.M. and S.R.; Methodology, D.S.; Software, N.S. and S.R.; Validation, U.M.; Formal analysis, N.M. and N.S.; Investigation, N.S.; Resources, D.S. and U.M.; Data curation, S.R.; Writing—original draft, D.S. and S.R.; Visualization, D.S.; Supervision, U.M. and N.M. All authors have read and agreed to the published version of the manuscript.

**Funding:** This research received no external funding.

**Data Availability Statement:** The data presented in this study are available on request from the corresponding author. The data are not publicly available due to privacy.

**Conflicts of Interest:** The authors declare no conflict of interest.

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
