# Peer review of "The Evaluation of Industry 5.0 Concepts: Social Network Analysis Approach"

_applsci, doi:10.3390/app14031291_

Round 1

Reviewer 1 Report

Comments and Suggestions for Authors

The topic of Industry 5.0 is up-to-date with a connection to social areas. The structure of the contribution meets the general requirements of IMRAD. The literature review describes the concept of Industry 5.0 in connection to servitization.

However, I found several errors, which decrease the quality level of contribution:

- In the second paragraph there are two identical sentences.

- In the description of relevance authors mentioned, that the topic of Industry 5.0 has been a research trend since 2022 by hundreds of authors, but they mentioned only ONE source.

- There are two identical paragraphs in chapters 2.4 (last) and 2.5 (last).

- The authors used a description of a small company with a number of employees 20-49, but the general limit is from 10. There is no description for the valuation.

- Authors also used individual industries according to NACE classification. However, all industries mentioned in Table 2 have different requirements in connection to Industry 4,0 and 5.0. There is no "logical" explanation of the choice.

- In Tables 3, 4, and 5, there are presented research results, but the explanations of these results are at a poor level.

After correction of the paper, it could be accepted for publication as an average one.

Author Response

Point by point answer to Reviewer 1 of the paper titled “The Evaluation of Industry 5.0 Concepts: Social Network Analysis Approach”

Comment 1: The topic of Industry 5.0 is up-to-date with a connection to social areas. The structure of the contribution meets the general requirements of IMRAD. The literature review describes the concept of Industry 5.0 in connection to servitization.

Answer: Thank you for your kind comment.

Comment 2: However, I found several errors, which decrease the quality level of contribution:

- In the second paragraph there are two identical sentences.

Answer: We express our gratitude to the reviewer for their valuable comment. In accordance with the suggestion, we have corrected this section and provided a new overview to gain a deeper understanding of the objectives and purpose of this study.

Comment 3: - In the description of relevance authors mentioned, that the topic of Industry 5.0 has been a research trend since 2022 by hundreds of authors, but they mentioned only ONE source.

Answer: We appreciate the suggestions about the sources; accordingly, we added more sources and described the development process of Industry 5.0 in the previous literature.

Comment 4: - There are two identical paragraphs in chapters 2.4 (last) and 2.5 (last).

Answer: Thank you for your input. We corrected this technical issue.

Comment 5: - The authors used a description of a small company with a number of employees 20-49, but the general limit is from 10. There is no description for the valuation. Authors also used individual industries according to NACE classification. However, all industries mentioned in Table 2 have different requirements in connection to Industry 4,0 and 5.0. There is no "logical" explanation of the choice

Answer: Based on the reviewers' comments, our objective is to demonstrate that the European Manufacturing Survey consortium employs a distinct selection of small, medium, and large companies based on their potential for innovation. Furthermore, we incorporate all NACE sectors to provide a comprehensive understanding of the implementation of Industry 5. In addressing the study's limitations, we have included a section suggesting that future research should concentrate on the NACE sector to deeply explain the phenomenon of Industry 5.0.

Comment 6: - In Tables 3, 4, and 5, there are presented research results, but the explanations of these results are at a poor level.

Answer: We are thankful to the reviewer for allowing us to strengthen the paper. According to the reviewer's comment, we tried to better explain every result from the tables, and we rewrote this part of the paper to be more clear and to enable additional value for the readers.

Comment 7: After correction of the paper, it could be accepted for publication as an average one.

Answer: Thank you for your kind comment. We have attempted to address all the suggestions provided by you.

We would like to express our deepest gratitude to reviewer for the given suggestions about improving our manuscript. For your convenience, we have highlighted in red the changes in the manuscript.

Reviewer 2 Report

Comments and Suggestions for Authors

Dear authors,

Congratulations for your efforts: I hope that you can see published your paper pretty soon. Please, accept the following commentaries as possible improvements:

a) Theoretical and methodological frameworks: 

Socialist schools were focused only in goods productions and material needs, so they cannot attend the developed capitalism, even less the talent capitalism and 5th industrial rev.

Mainstream: new-Keynesians defend the green growth, and post-Keynesians defend the degrowth; both they defend the market failures... maybe they don´t understand the digital economy and the new labor relations and business culture.

Heterodox: maybe, the Austrian Economics can support your research, because they defend the dynamic process (Mises, 1949; Hayek, 1988; Huerta de Soto, 2009), with decentralized and cooperative relations (idem). They defend the creativity and the entrepreneurship. Also, you can reinforce this framework with new-Institutional Approaches (Law & Economics: Coase, Posner, Calabresi, etc.; Public Choice: Buchanan, Tullock, Breenan, etc.). They attend the efficiency and the quality development of the institutions and the economic system. An example to illustrate the collaboration between Austrian Economics and new-Institutionalism: https://doi.org/10.3389/fpubh.2022.801525

So, do you think possible to connect your research question with the following thougths: a) from manufacture to "mindfacture" (main production factor is knowledge and talent); b) readjustment effect (from welfare state economy -interventionism to provide material goods- to wellbeing economics -tech connects the economic agents, to provide material and inmaterial neccesities), etc. So, resilience to keep doing the things into the welfare state economy or resilience to improve the wellbeing economics.

Layout: 

Keywords: end with period (not ;)

Tables: source? (own elaboration?)

Graphs: source? (idem).

References: are you sure that´s style? [7]: pages?

Again, congratulations and best regards.

Comments on the Quality of English Language

It´s ok

Author Response

Point by point answer to Reviewer 2 of the paper titled “The Evaluation of Industry 5.0 Concepts: Social Network Analysis Approach”

Comment 1: Dear authors,

Congratulations for your efforts: I hope that you can see published your paper pretty soon.

Answer: Thank you for your kind comment.

Comment 2: Please, accept the following commentaries as possible improvements:

  1. a) Theoretical and methodological frameworks:

Socialist schools were focused only in goods productions and material needs, so they cannot attend the developed capitalism, even less the talent capitalism and 5th industrial rev.

Mainstream: new-Keynesians defend the green growth, and post-Keynesians defend the degrowth; both they defend the market failures... maybe they don´t understand the digital economy and the new labor relations and business culture.

Heterodox: maybe, the Austrian Economics can support your research, because they defend the dynamic process (Mises, 1949; Hayek, 1988; Huerta de Soto, 2009), with decentralized and cooperative relations (idem). They defend the creativity and the entrepreneurship. Also, you can reinforce this framework with new-Institutional Approaches (Law & Economics: Coase, Posner, Calabresi, etc.; Public Choice: Buchanan, Tullock, Breenan, etc.). They attend the efficiency and the quality development of the institutions and the economic system. An example to illustrate the collaboration between Austrian Economics and new-Institutionalism: https://doi.org/10.3389/fpubh.2022.801525

So, do you think possible to connect your research question with the following thougths: a) from manufacture to "mindfacture" (main production factor is knowledge and talent); b) readjustment effect (from welfare state economy -interventionism to provide material goods- to wellbeing economics -tech connects the economic agents, to provide material and inmaterial neccesities), etc. So, resilience to keep doing the things into the welfare state economy or resilience to improve the wellbeing economics.

Answer: We express gratitude to the reviewer for affording us the opportunity to enhance the article. In response to the reviewer's comments, we aim to elucidate the rationale behind Industry 5.0 in the literature section, specifically focusing on Human-Centricity as suggested by the reviewer. Furthermore, in the section on Theoretical Implications, we have incorporated a paragraph that strives to provide a more comprehensive explanation of the phenomena outlined in your comment and how our study contributes to it.

Comment 3: Layout:

Keywords: end with period (not ;)

Answer: Thank you for this comment. We fixed this technical error.

Comment 4: Tables and Graph: source? (own elaboration?)

Answer: Thank you for your question. The tables and graphs were created by ourselves.

Comment 6: References: are you sure that´s style? [7]: pages?

Answer: Thank you for this comment. We fixed this technical error.

Comment 7: Again, congratulations and best regards.

Answer: We are sincerely thankful!

We would like to express our deepest gratitude to reviewer for the given suggestions about improving our manuscript. For your convenience, we have highlighted in red the changes in the manuscript.

Reviewer 3 Report

Comments and Suggestions for Authors

Thank you for the opportunity to review the manuscript (applsci-2834416) titled "The Evaluation of Industry 5.0 Concepts: Social Network Analysis Approach." I have the following comments:

Abstract:

Please discuss the contribution of the study in the last few lines.

Introduction:

The introduction is about motivation, gaps, and contributions. None of them are adequately explained.

  • The rationale behind this research needs clearer motivation, necessitating a more thorough explanation of its significance. The arguments presented in the initial paragraph are somewhat unsubstantial and lack persuasiveness. Consequently, a few stronger arguments are essential to effectively elucidating the study's motivation.
  • Social development goals or sustainable development goals?
  • The gap seems vague and unsubstantial. Is there no study available on Industry 5.0 in developing countries? If there are studies available, then this gap analysis does not make any sense. What is the problem or gap in previous studies must be discussed in detail? What is new should be explained more clearly.
  • A paragraph on the study's contribution is also required.

Literature Review, Methodology, Results, and Conclusion

  • The literature is appropriate, adequate, and up-to-date. All approaches are explained in a good manner.
  • Why is Serbia an important context? Add some facts and figures about manufacturing firms in the Siberian economy.
  • The methods employed are suitable. The model appears to be accurately constructed and effectively presented, with clear explanations provided for each element.
  • The chosen analytical approach in this study is deemed robust, and the results are interesting. They are properly and correctly interpreted.
  • There is a need to compare the results with previous studies, which could help us understand the difference between this research and the previous one.
  • The practical implications are very well explained. However, what is the contribution to the literature? Theoretical implications are missing.
  • The limitations and future research directions are not properly explained. The limitations should be acknowledged, and clear future research directions are required.

Additional comments

·       Proofreading is recommended for removing grammatical errors and improving readability.

Comments on the Quality of English Language

The language is hard to follow. Professional proofreading is recommended.

Author Response

Point by point answer to Reviewer 3 of the paper titled “The Evaluation of Industry 5.0 Concepts: Social Network Analysis Approach”

Comment 1: Thank you for the opportunity to review the manuscript (applsci-2834416) titled "The Evaluation of Industry 5.0 Concepts: Social Network Analysis Approach."

Answer: Thank you for reviewing the manuscript and for giving us the opportunity to strengthen the paper.

Comment 2: I have the following comments:

Abstract: Please discuss the contribution of the study in the last few lines.

Answer: Thank you for these kind comments. We integrate the main findings of your study into the abstract.

Comment 3: Introduction:

The introduction is about motivation, gaps, and contributions. None of them are adequately explained.

The rationale behind this research needs clearer motivation, necessitating a more thorough explanation of its significance. The arguments presented in the initial paragraph are somewhat unsubstantial and lack persuasiveness. Consequently, a few stronger arguments are essential to effectively elucidating the study's motivation.

Answer: Thank you for your comments, which have facilitated the improvement of our manuscript. In the introduction, we delve into the explanation of certain concepts and address the gaps in previous research on Industry 5.0. We have also tried to provide a clearer explanation of the main needs and aims of our study.

Comment 4: Social development goals or sustainable development goals?

Answer: Thank you for this comment. We fixed this technical error. This are Sustainable development goals

Comment 5: The gap seems vague and unsubstantial. Is there no study available on Industry 5.0 in developing countries? If there are studies available, then this gap analysis does not make any sense. What is the problem or gap in previous studies must be discussed in detail? What is new should be explained more clearly.

A paragraph on the study's contribution is also required.

Answer: The reviewer's feedback is highly valued. In the revised version, we clarify that previous research predominantly focuses on theoretical concepts of Industry 5.0 in developing countries. However, our study introduces novel results and insights from the European Manufacturing Survey to provide a more comprehensive understanding of the main gaps. This serves as our primary contribution to both the scientific and practical environments.

Comment 6: Literature Review, Methodology, Results, and Conclusion

The literature is appropriate, adequate, and up-to-date. All approaches are explained in a good manner.

Answer: Thank you for your kind comment.

Comment 7: Why is Serbia an important context? Add some facts and figures about manufacturing firms in the Siberian economy.

Answer: We are thankful to the reviewer for this comment. The authors chose the Republic of Serbia to represent developing countries due to previous literature in the manufacturing sector identifying this country as a notable example within the research consortium of the European Manufacturing Survey. Accordingly, we mentioned previous research in the literature which shows the main concept of the manufacturing sector in previous research.

Comment 8: The methods employed are suitable. The model appears to be accurately constructed and effectively presented, with clear explanations provided for each element.

The chosen analytical approach in this study is deemed robust, and the results are interesting. They are properly and correctly interpreted.

Answer: We are very grateful for this comment!

Comment 9: There is a need to compare the results with previous studies, which could help us understand the difference between this research and the previous one. The practical implications are very well explained. However, what is the contribution to the literature? Theoretical implications are missing.

Answer: Thank you for your valuable comment. We have added a new paragraph in the Theoretical Implications section that provides a better explanation of how the results from our study fill the gap in the literature and confirm previous findings.

Comment 11: The limitations and future research directions are not properly explained. The limitations should be acknowledged, and clear future research directions are required.

Answer: We are thankful to the reviewer for this comment. We have incorporated a new section in the Discussion that explains the primary limitations and future implications of our study.

Comment 12: Additional comments

Proofreading is recommended for removing grammatical errors and improving readability.

Answer: Thank you for the comment. The manuscript has been proofread by a native speaker. We hope that it now appears easier for reading.

We would like to express our deepest gratitude to reviewer for the given suggestions about improving our manuscript. For your convenience, we have highlighted in red the changes in the manuscript.

Round 2

Reviewer 3 Report

Comments and Suggestions for Authors

Thank you for the opportunity to review the revised manuscript (applsci-2834416) titled "The Evaluation of Industry 5.0 Concepts: Social Network Analysis Approach." I enjoyed reading this revised manuscript. I found the revised manuscript to be a substantial improvement. The author(s) diligently addressed most of my inquiries and concerns, including an improved abstract and an emphasis on research gaps. The presentation and comparison of results with previous studies are well executed. The implications are thoroughly discussed, and the limitations and future research directions are clearly outlined. However, some comments are not sufficiently addressed, such as

·       The study motivation needs to be improved by answering why we should care about this research.

·       I could not find a contribution paragraph in the introduction section after the research questions.

·       In the methodology, when discussing the context, add facts and figures (contributions) about manufacturing firms in the Siberian economy.

Best of luck with your work.

Author Response

Point by point answer to Reviewer 3 of the paper titled “The Evaluation of Industry 5.0 Concepts: Social Network Analysis Approach” – 2nd review round

Comment 1: Thank you for the opportunity to review the revised manuscript (applsci-2834416) titled "The Evaluation of Industry 5.0 Concepts: Social Network Analysis Approach." I enjoyed reading this revised manuscript. I found the revised manuscript to be a substantial improvement. The author(s) diligently addressed most of my inquiries and concerns, including an improved abstract and an emphasis on research gaps. The presentation and comparison of results with previous studies are well executed. The implications are thoroughly discussed, and the limitations and future research directions are clearly outlined.

Answer: Thank you for reviewing the revised manuscript. We found the comments very insightful and helpful with strengthening the paper.

Comment 2: However, some comments are not sufficiently addressed, such as

The study motivation needs to be improved by answering why we should care about this research.

Answer: Thank you for your comment. In the introduction, we emphasized the importance of this research by explaining which industries have and which haven’t been covered with potential Industry 5.0 applications, and stating that findings from the manufacturing sector are lacking.

Comment 3: I could not find a contribution paragraph in the introduction section after the research questions.

Answer: The reviewer's feedback is highly valued. We have added a paragraph after the research questions in the introduction which gives insights about the contribution of this paper.

Comment 4: In the methodology, when discussing the context, add facts and figures (contributions) about manufacturing firms in the Siberian economy.

Answer: We are thankful to the reviewer for this comment. We have added facts about the Serbian population and the sample used in the conducted research. The facts cover the total number of firms, how are they grouped by firm sized, which manufacturing sectors are the most common, and which Serbian districts are most common, when talking about manufacturing firms.

We would like to express our deepest gratitude to reviewer for the given suggestions about improving our manuscript. For your convenience, we have highlighted in red the changes in the manuscript.